# Multi-Task Bayesian Optimization

**Kevin Swersky**
Department of Computer Science
University of Toronto
kswersky@cs.toronto.edu

**Jasper Snoek**[*]
School of Engineering and Applied Sciences
Harvard University
jsnoek@seas.harvard.edu

**Ryan P. Adams**
School of Engineering and Applied Sciences
Harvard University
rpa@seas.harvard.edu

## Abstract

Bayesian optimization has recently been proposed as a framework for automatically tuning the hyperparameters of machine learning models and has been shown to yield state-of-the-art performance with impressive ease and efficiency. In this paper, we explore whether it is possible to transfer the knowledge gained from previous optimizations to new tasks in order to find optimal hyperparameter settings more efficiently. Our approach is based on extending multi-task Gaussian processes to the framework of Bayesian optimization. We show that this method significantly speeds up the optimization process when compared to the standard single-task approach. We further propose a straightforward extension of our algorithm in order to jointly minimize the average error across multiple tasks and demonstrate how this can be used to greatly speed up $k$-fold cross-validation. Lastly, we propose an adaptation of a recently developed acquisition function, entropy search, to the cost-sensitive, multi-task setting. We demonstrate the utility of this new acquisition function by leveraging a small dataset to explore hyperparameter settings for a large dataset. Our algorithm dynamically chooses which dataset to query in order to yield the most information per unit cost.

## 1 Introduction

The proper setting of high-level hyperparameters in machine learning algorithms – regularization weights, learning rates, etc. – is crucial for successful generalization. The difference between poor settings and good settings of hyperparameters can be the difference between a useless model and state-of-the-art performance. Surprisingly, hyperparameters are often treated as secondary considerations and are not set in a documented and repeatable way. As the field matures, machine learning models are becoming more complex, leading to an increase in the number of hyperparameters, which often interact with each other in non-trivial ways. As the space of hyperparameters grows, the task of tuning them can become daunting, as well-established techniques such as grid search either become too slow, or too coarse, leading to poor results in both performance and training time.

Recent work in machine learning has revisited the idea of *Bayesian optimization* [1, 2, 3, 4, 5, 6, 7], a framework for global optimization that provides an appealing approach to the difficult exploration-exploitation tradeoff. These techniques have been shown to obtain excellent performance on a variety of models, while remaining efficient in terms of the number of required function evaluations, corresponding to the number of times a model needs to be trained.

One issue with Bayesian optimization is the so-called "cold start" problem. The optimization must be carried out from scratch each time a model is applied to new data. If a model will be applied to

---

[*]Research was performed while at the University of Toronto.

many different datasets, or even just a few extremely large datasets, then there may be a significant overhead to re-exploring the same hyperparameter space. Machine learning researchers are often faced with this problem, and one appealing solution is to transfer knowledge from one domain to the next. This could manifest itself in many ways, including establishing the values for a grid search, or simply taking certain hyperparameters as fixed with some commonly accepted value. Indeed, it is this knowledge that often separates an expert machine learning practitioner from a novice.

The question that this paper explores is whether we can incorporate the same kind of transfer of knowledge within the Bayesian optimization framework. Such a tool would allow researchers and practitioners to leverage previously trained models in order to quickly tune new ones. Furthermore, for large datasets one could imagine exploring a wide range of hyperparameters on a small subset of data, and then using this knowledge to quickly find an effective setting on the full dataset with just a few function evaluations.

In this paper, we propose multi-task Bayesian optimization to solve this problem. The basis for the idea is to apply well-studied multi-task Gaussian process models to the Bayesian optimization framework. By treating new domains as new tasks, we can adaptively learn the degree of correlation between domains and use this information to hone the search algorithm. We demonstrate the utility of this approach in a number of different settings: using prior optimization runs to bootstrap new runs; optimizing multiple tasks simultaneously when the goal is maximizing average performance; and utilizing a small version of a dataset to explore hyperparameter settings for the full dataset. Our approach is fully automatic, requires minimal human intervention and yields substantial improvements in terms of the speed of optimization.

## 2 Background

### 2.1 Gaussian Processes

Gaussian processes (GPs) [8] are a flexible class of models for specifying prior distributions over functions $f : \mathcal{X} \to \mathbb{R}$. They are defined by the property that any finite set of $N$ points $\mathbf{X} = \{\mathbf{x}_n \in \mathcal{X}\}_{n=1}^{N}$ induces a Gaussian distribution on $\mathbb{R}^N$. The convenient properties of the Gaussian distribution allow us to compute marginal and conditional means and variances in closed form. GPs are specified by a mean function $m : \mathcal{X} \to \mathbb{R}$ and a positive definite covariance, or kernel function $K : \mathcal{X} \times \mathcal{X} \to \mathbb{R}$. The predictive mean and covariance under a GP can be respectively expressed as:

$$\mu(\mathbf{x}\,;\,\{\mathbf{x}_n, y_n\}, \theta) = K(\mathbf{X}, \mathbf{x})^{\top} K(\mathbf{X}, \mathbf{X})^{-1}(\mathbf{y} - m(\mathbf{X})), \tag{1}$$

$$\Sigma(\mathbf{x}, \mathbf{x}'\,;\,\{\mathbf{x}_n, y_n\}, \theta) = K(\mathbf{x}, \mathbf{x}') - K(\mathbf{X}, \mathbf{x})^{\top} K(\mathbf{X}, \mathbf{X})^{-1} K(\mathbf{X}, \mathbf{x}'). \tag{2}$$

Here $K(\mathbf{X}, \mathbf{x})$ is the $N$-dimensional column vector of cross-covariances between $\mathbf{x}$ and the set $\mathbf{X}$. The $N \times N$ matrix $K(\mathbf{X}, \mathbf{X})$ is the Gram matrix for the set $\mathbf{X}$. As in [6] we use the Matérn $5/2$ kernel and we marginalize over kernel parameters $\theta$ using slice sampling [9].

### 2.2 Multi-Task Gaussian Processes

In the field of geostatistics [10, 11], and more recently in the field of machine learning [12, 13, 14], Gaussian processes have been extended to the case of vector-valued functions, i.e., $f : \mathcal{X} \to \mathbb{R}^T$. We can interpret the $T$ outputs of such functions as belonging to different regression tasks. The key to modeling such functions with Gaussian processes is to define a useful covariance function $K((\mathbf{x}, t), (\mathbf{x}', t'))$ between input-task pairs. One simple approach is called the *intrinsic model of coregionalization* [12, 11, 13], which transforms a latent function to produce each output. Formally,

$$K_{\mathsf{multi}}((\mathbf{x}, t), (\mathbf{x}', t')) = K_{\mathsf{t}}(t, t') \otimes K_{\mathsf{x}}(\mathbf{x}, \mathbf{x}'), \tag{3}$$

where $\otimes$ denotes the Kronecker product, $K_{\mathsf{x}}$ measures the relationship between inputs, and $K_{\mathsf{t}}$ measures the relationship between tasks. Given $K_{\mathsf{multi}}$, this is simply a standard GP. Therefore, the complexity still grows cubically in the total number of observations.

Along with the other kernel parameters, we infer the parameters of $K_{\mathsf{t}}$ using slice sampling. Specifically, we represent $K_{\mathsf{t}}$ by its Cholesky factor and sample in that space. For our purposes, it is reasonable to assume a positive correlation between tasks. We found that sampling each element of the Cholesky in log space and then exponentiating adequately satisfied this constraint.

## 2.3 Bayesian Optimization for a Single Task

Bayesian optimization is a general framework for the global optimization of noisy, expensive, black-box functions [15]. The strategy is based on the notion that one can use a relatively cheap probabilistic model to query as a surrogate for the financially, computationally or physically expensive function that is subject to the optimization. Bayes' rule is used to derive the posterior estimate of the true function given observations, and the surrogate is then used to determine the next most promising point to query. A common approach is to use a GP to define a distribution over objective functions from the input space to a loss that one wishes to minimize. That is, given observation pairs of the form $\{\mathbf{x}_n, y_n\}_{n=1}^N$, where $\mathbf{x}_n \in \mathcal{X}$ and $y_n \in \mathbb{R}$, we assume that the function $f(\mathbf{x})$ is drawn from a Gaussian process prior where $y_n \sim \mathcal{N}(f(\mathbf{x}_n), \nu)$ and $\nu$ is the function observation noise variance.

A standard approach is to select the next point to query by finding the maximum of an *acquisition function* $a(\mathbf{x}\,;\,\{\mathbf{x}_n, y_n\}, \theta)$ over a bounded domain in $\mathcal{X}$. This is an heuristic function that uses the posterior mean and uncertainty, conditioned on the GP hyperparameters $\theta$, in order to balance exploration and exploitation. There have been many proposals for acquisition functions, or combinations thereof [16, 2]. We will use the expected improvement criterion (EI) [15, 17],

$$a_{\mathsf{EI}}(\mathbf{x}\,;\,\{\mathbf{x}_n, y_n\}, \theta) = \sqrt{\Sigma(\mathbf{x}, \mathbf{x}\,;\,\{\mathbf{x}_n, y_n\}, \theta)}\,(\gamma(\mathbf{x})\,\Phi(\gamma(\mathbf{x})) + \mathcal{N}(\gamma(\mathbf{x})\,;\,0, 1))\,, \qquad (4)$$

$$\gamma(\mathbf{x}) = \frac{y_{\text{best}} - \mu(\mathbf{x}\,;\,\{\mathbf{x}_n, y_n\}, \theta)}{\sqrt{\Sigma(\mathbf{x}, \mathbf{x}\,;\,\{\mathbf{x}_n, y_n\}, \theta)}}. \qquad (5)$$

Where $\Phi(\cdot)$ is the cumulative distribution function of the standard normal, and $\gamma(\mathbf{x})$ is a $Z$-score. Due to its simple form, EI can be locally optimized using standard black-box optimization algorithms [6].

An alternative to heuristic acquisition functions such as EI is to consider a distribution over the minimum of the function and iteratively evaluating points that will most decrease the entropy of this distribution. This *entropy search* strategy [18] has the appealing interpretation of decreasing the uncertainty over the location of the minimum at each optimization step. Here, we formulate the entropy search problem as that of selecting the next point from a pre-specified candidate set. Given a set of $C$ points $\tilde{\mathbf{X}} \subset \mathcal{X}$, we can write the probability of a point $\mathbf{x} \in \tilde{\mathbf{X}}$ having the minimum function value among the points in $\tilde{\mathbf{X}}$ via:

$$\Pr(\text{min at } \mathbf{x}\,|\,\theta, \tilde{\mathbf{X}}, \{\mathbf{x}_n, y_n\}_{n=1}^N) = \int_{\mathbb{R}^C} p(\mathbf{f}\,|\,\mathbf{x}, \theta, \{\mathbf{x}_n, y_n\}_{n=1}^N) \prod_{\tilde{\mathbf{x}} \in \tilde{\mathbf{X}} \backslash \mathbf{x}} h\,(f(\tilde{\mathbf{x}}) - f(\mathbf{x}))\,\mathrm{d}\mathbf{f}, \quad (6)$$

where $\mathbf{f}$ is the vector of function values at the points $\tilde{\mathbf{X}}$ and $h$ is the Heaviside step function. The entropy search procedure relies on an estimate of the reduction in uncertainty over this distribution if the value $y$ at $\mathbf{x}$ is revealed. Writing $\Pr(\text{min at } \mathbf{x}\,|\,\theta, \tilde{\mathbf{X}}, \{\mathbf{x}_n, y_n\}_{n=1}^N)$ as $\mathrm{P}_{\mathsf{min}}$, $p(\mathbf{f}\,|\,\mathbf{x}, \theta, \{\mathbf{x}_n, y_n\}_{n=1}^N)$ as $p(\mathbf{f}\,|\,\mathbf{x})$ and the GP likelihood function as $p(y\,|\,\mathbf{f})$ for brevity, and using $H(\mathrm{P})$ to denote the entropy of $\mathrm{P}$, the objective is to find the point $\mathbf{x}$ from a set of candidates which maximizes the *information gain* over the distribution of the location of the minimum,

$$a_{\mathsf{KL}}(\mathbf{x}) = \int \int [H(\mathrm{P}_{\mathsf{min}}) - H(\mathrm{P}_{\mathsf{min}}^y)]\,p(y\,|\,\mathbf{f})\,p(\mathbf{f}\,|\,\mathbf{x})\,\mathrm{d}y\,\mathrm{d}\mathbf{f}, \qquad (7)$$

where $\mathrm{P}_{\mathsf{min}}^y$ indicates that the fantasized observation $\{\mathbf{x}, y\}$ has been added to the observation set. Although (7) does not have a simple form, we can use Monte Carlo to approximate it by sampling $\mathbf{f}$. An alternative to this formulation is to consider the reduction in entropy relative to a uniform base distribution, however we found that the formulation given by Equation (7) works better in practice.

# 3 Multi-Task Bayesian Optimization

## 3.1 Transferring Bayesian Optimization to a New Task

Under the framework of multi-task GPs, performing optimization on a related task is fairly straightforward. We simply restrict our future observations to the task of interest and proceed as normal. Once we have enough observations on the task of interest to properly estimate $K_{\mathrm{t}}$, then the other tasks will act as additional observations without requiring any additional function evaluations. An illustration of a multi-task GP versus a single-task GP and its effect on EI is given in Figure 1. This approach can be thought of as a special case of contextual Gaussian process bandits [19].

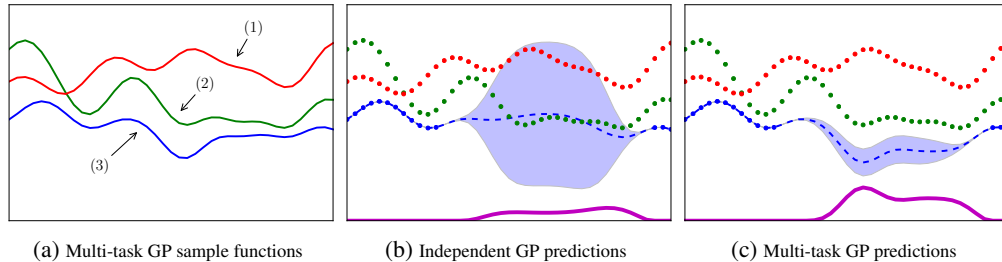

|  (a) Multi-task GP sample functions | (b) Independent GP predictions | (c) Multi-task GP predictions |

Figure 1: (a) A sample function with three tasks from a multi-task GP. Tasks 2 and 3 are correlated, 1 and 3 are anti-correlated, and 1 and 2 are uncorrelated. (b) independent and (c) multi-task predictions on the third task. The dots represent observations, while the dashed line represents the predictive mean. Here we show a function over three tasks and corresponding observations, where the goal is to minimize the function over the third task. The curve shown on the bottom represents the expected improvement for each input location on this task. The independent GP fails to adequately represent the function and optimizing EI leads to a spurious evaluation. The multi-task GP utilizes the other tasks and the maximal EI point corresponds to the true minimum.

## 3.2 Optimizing an Average Function over Multiple Tasks

Here we will consider optimizing the average function over multiple tasks. This has elements of both single and multi-task settings since we have a single objective representing a joint function over multiple tasks. We motivate this approach by considering a finer-grained version of Bayesian optimization over $k$-fold cross validation. We wish to optimize the average performance over all $k$ folds, but it may not be necessary to actually evaluate all of them in order to identify the quality of the hyperparameters under consideration. The predictive mean and variance of the average objective are given by:

$$\bar{\mu}(\mathbf{x}) = \frac{1}{k} \sum_{t=1}^{k} \mu(\mathbf{x}, t\,; \{\mathbf{x}_n, y_n\}, \theta), \qquad \bar{\sigma}(\mathbf{x})^2 = \frac{1}{k^2} \sum_{t=1}^{k} \sum_{t'=1}^{k} \Sigma(\mathbf{x}, \mathbf{x}, t, t'\,; \{\mathbf{x}_n, y_n\}, \theta). \qquad (8)$$

If we are willing to spend one function evaluation on each task for every point $\mathbf{x}$ that we query, then the optimization of this objective can proceed using standard approaches. In many situations though, this can be expensive and perhaps even wasteful. As an extreme case, if we have two perfectly correlated tasks then spending two function evaluations per query provides no additional information, at twice the cost of a single-task optimization. The more interesting case then is to try to jointly choose both $\mathbf{x}$ as well as the task $t$ and spend only one function evaluation per query.

We choose a $(\mathbf{x}, t)$ pair using a two-step heuristic. First we impute missing observations using the predictive means. We then use the estimated average function to pick a promising candidate $\mathbf{x}$ by optimizing EI. Conditioned on $\mathbf{x}$, we then choose the task that yields the highest single-task expected improvement.

The problem of minimizing the average error over multiple tasks has been considered in [20], where they applied Bayesian optimization in order to tune a single model on multiple datasets. Their approach is to project each function to a joint latent space and then iteratively visit each dataset in turn. Another approach can be found in [3], where additional task-specific features are used in conjunction with the inputs $\mathbf{x}$ to make predictions about each task.

## 3.3 A Principled Multi-Task Acquisition Function

Rather than transferring knowledge from an already completed search on a related task to bootstrap a new one, a more desirable strategy would have the optimization routine dynamically query the related, possibly significantly cheaper task. Intuitively, if two tasks are closely related, then evaluating a cheaper one can reveal information and reduce uncertainty about the location of the minimum on the more expensive task. A clever strategy may, for example, perform low cost exploration of a promising location on the cheaper task before risking an evaluation of the expensive task. In this section we develop an acquisition function for such a dynamic multi-task strategy which specifically takes noisy estimates of *cost* into account based on the entropy search strategy.

Although the EI criterion is intuitive and effective in the single task case, it does not directly generalize to the multi-task case. However, entropy search does translate naturally to the multi-task problem. In this setting we have observation pairs from multiple tasks, $\{\mathbf{x}_n^t, y_n^t\}_{n=1}^N$ and we wish

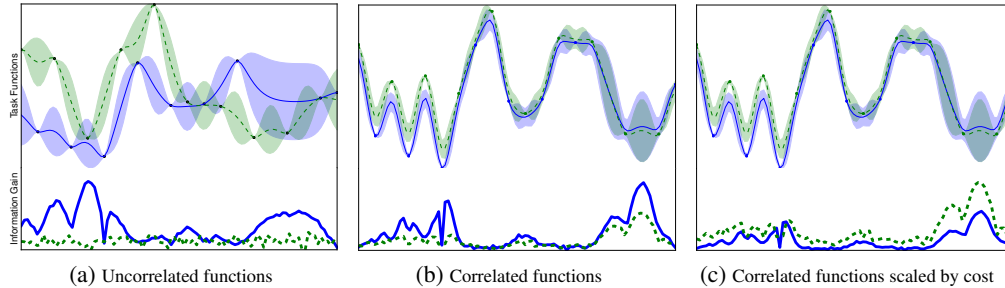

(a) Uncorrelated functions          (b) Correlated functions          (c) Correlated functions scaled by cost

Figure 2: A visualization of the multi-task information gain per unit cost acquisition function. In each figure, the objective is to find the minimum of the solid blue function. The green function is an auxiliary objective function. In the bottom of each figure are lines indicating the expected information gain with regard to the primary objective function. The green dashed line shows the information gain about the *primary* objective that results from evaluating the *auxiliary* objective function. Figure 2a shows two sampled functions from a GP that are uncorrelated. Evaluating the primary objective gains information, but evaluating the auxiliary does not. In Figure 2b we see that with two strongly correlated functions, not only do observations on either task reduce uncertainty about the other, but observations from the auxiliary task acquire information about the primary task. Finally, in 2c we assume that the primary objective is three times more expensive than the auxiliary task and thus evaluating the related task gives *more* information gain per unit cost.

to pick the candidate $\mathbf{x}^t$ that maximally reduces the entropy of $\mathrm{P}_{\mathsf{min}}$ for the primary task, which we take to be $t = 1$. Naturally, $\mathrm{P}_{\mathsf{min}}$ evaluates to zero for $\mathbf{x}^{t>1}$. However, we can evaluate $\mathrm{P}_{\mathsf{min}}^y$ for $y^{t>1}$ and if the auxiliary task is related to the primary task, $\mathrm{P}_{\mathsf{min}}^y$ will change from the base distribution and $H(\mathrm{P}_{\mathsf{min}}) - H(\mathrm{P}_{\mathsf{min}}^y)$ will be positive. Through reducing uncertainty about f, evaluating an observation on a related auxiliary task can reduce the entropy of $\mathrm{P}_{\mathsf{min}}$ on the primary task of interest.

However, observe that evaluating a point on a related task can never reveal *more* information than evaluating the same point on the task of interest. Thus, the above strategy would never choose to evaluate a related task. Nevertheless, when cost is taken into account, the auxiliary task may convey more information per unit cost. Thus we translate the objective from Equation (7) to instead reflect the *information gain per unit cost* of evaluating a candidate point,

$$a_{\mathsf{IG}}(\mathbf{x}^t) = \int \int \left( \frac{H[\mathrm{P}_{\mathsf{min}}] - H[\mathrm{P}_{\mathsf{min}}^y]}{c_t(\mathbf{x})} \right) p(y \mid \mathrm{f}) \, p(\mathrm{f} \mid \mathbf{x}^t) \, \mathrm{d}y \, \mathrm{d}\mathrm{f}, \qquad (9)$$

where $c_t(\mathbf{x})$, $c_t : \mathcal{X} \to \mathbb{R}^+$, is the real valued cost of evaluating task $t$ at $\mathbf{x}$. Although, we don't know this cost function in advance, we can estimate it similarly to the task functions, $f(\mathbf{x}^t)$, using the same multi-task GP machinery to model $\log c_t(\mathbf{x})$.

Figure 2 provides a visualization of this acquisition function, using a two task example. It shows how selecting a point on a related auxiliary task can reduce uncertainty about the location of the minimum on the primary task of interest (blue solid line). In this paper, we assume that all the candidate points for which we compute $a_{\mathsf{IG}}$ come from a fixed subset. Following [18], we pick these candidates by taking the top $C$ points according to the EI criterion on the primary task of interest.

## 4   Empirical Analyses

### 4.1   Addressing the Cold Start Problem

Here we compare Bayesian optimization with no initial information to the case where we can leverage results from an already completed optimization on a related task. In each classification experiment the target of Bayesian optimization is the error on a held out validation set. Further details on these experiments can be found in the supplementary material.

**Branin-Hoo**   The Branin-Hoo function is a common benchmark for optimization techniques [17] that is defined over a bounded set on $\mathbb{R}^2$. As a related task we consider a shifted Branin-Hoo where the function is translated by 10% along either axis. We used Bayesian optimization to find the minimum of the original function and then added the shifted function as an additional task.

**Logistic regression**   We optimize four hyperparameters of logistic regression (LR) on the MNIST dataset using 10000 validation examples. We assume that we have already completed 50 iterations

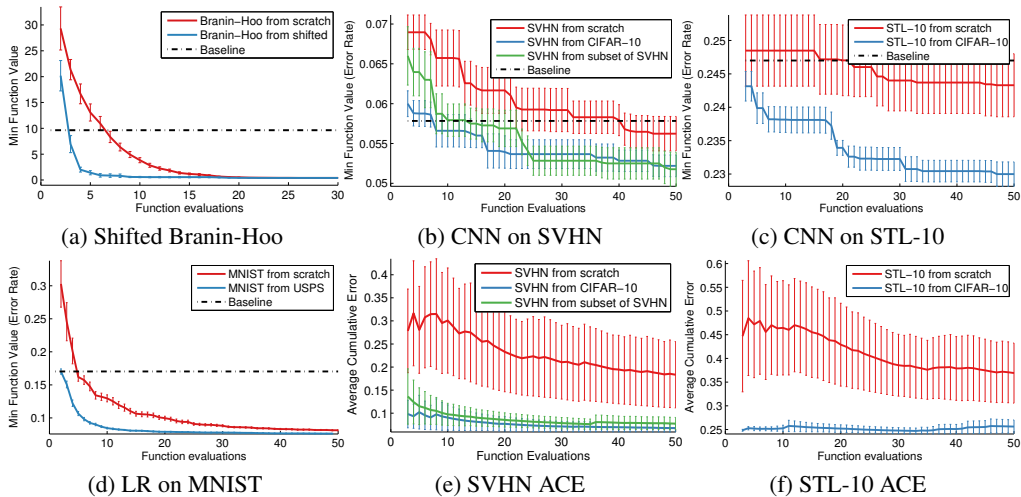

Figure 3: (a)-(d)Validation error per function evaluation. (e),(f) ACE over function evaluations.

of an optimization of the same model on the related USPS digits task. The USPS data is only 1/6 the size of MNIST and each image contains $16 \times 16$ pixels, so it is considerably cheaper to evaluate.

**Convolutional neural networks on pixels** We applied convolutional neural networks[1] (CNNs) to the Street View House Numbers (SVHN) [21] dataset and bootstrapped from a previous run of Bayesian optimization using the same model trained on CIFAR-10 [22, 6]. At the time, this model represented the state-of-the-art. The SVHN dataset has the same input dimension as CIFAR-10, but is 10 times larger. We used 6000 held-out examples for validation. Additionally, we consider training on 1/10th of the SVHN dataset to warm-start the full optimization. The best settings yielded $4.77 \pm 0.22\%$ error, which is comparable to domain experts using non-dropout CNNs [23].

**Convolutional networks on k-means features** As an extension to the previous CNN experiment, we incorporate a more sophisticated pipeline in order to learn a model for the STL-10 dataset [24]. This dataset consists of images with $96 \times 96$ pixels, and each training set has only 1000 images. Overfitting is a significant challenge for this dataset, so we utilize a CNN[2] on top of k-means features in a similar approach to [25], as well as dropout [26]. We bootstrapped Bayesian optimization using the same model trained on CIFAR-10, which had achieved $14.2\%$ test error on that dataset. During the optimization, we used the first fold for training, and the remaining 4000 points from the other folds for validation. We then trained separate networks on each fold using the best hyperparameter settings found by Bayesian optimization. Following reporting conventions for this dataset, the model achieved $70.1 \pm 0.6\%$ test-set accuracy, exceeding the previous state-of-the-art of $64.5 \pm 1\%$ [27].

The results of these experiments are shown in Figure 3(a)-(d). In each case, the multi-task optimization finds a better function value much more quickly than single-task optimization. Clearly there is information in the related tasks that can be exploited. To better understand the behaviour of the different methods, we plot the average cumulative error (ACE), i.e., the average of all function values seen up to a given time, in Figure 3(e),(f). The single-task method wastes many more evaluations exploring poor hyperparameter settings. In the multi-task case, this exploration has already been performed and more evaluations are spent on exploitation.

As a baseline (the dashed black line), we took the best model from the first task and applied it directly to the task of interest. For example, in the CNN experiments this involved taking the best settings from CIFAR-10. This "direct transfer" performed well in some cases and poorly in others. In general, we have found that the best settings for one task are usually not optimal for the other.

## 4.2 Fast Cross-Validation

$k$-fold cross-validation is a widely used technique for estimating the generalization error of machine learning models, but requires retraining a model $k$ times. This can be prohibitively expensive with complex models and large datasets. It is reasonable to expect, however, that if the data are randomly

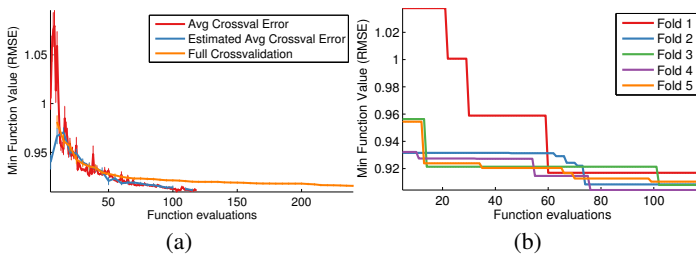

Figure 4: (a) PMF cross-validation error per function evaluation on Movielens-100k. (b) Lowest error observed for each fold per function evaluation for a single run.

partitioned among folds that the errors for each fold will be highly correlated. For a given set of hyperparameters, we can therefore expect diminishing returns in estimating the average error for each subsequently evaluated fold. With a good GP model, we can very likely obtain a high quality estimate by evaluating just one fold per setting. In this experiment, we apply the algorithm described in Section 3.2 in order to dynamically determine which points/folds to query.

We demonstrate this procedure on the task of training probabilistic matrix factorization (PMF) models for recommender systems [28]. The hyperparameters of the PMF model are the learning rate, an $\ell_2$ regularizer, the matrix rank, and the number of epochs. We use 5-fold cross validation on the Movielens-100k dataset [29]. In Figure 4(a) we show the best error obtained after a given number of function evaluations as measured by the number of folds queried, averaged over 50 optimization runs. For the multi-task version, we show both the true average cross-validation error, as well as the estimated error according to the GP. In the beginning, the GP fit is highly uncertain, so the optimization exhibits some noise. As the GP model becomes more certain however, the true error and the GP estimate converge and the search proceeds rapidly compared to the single-task counterpart. In Figure 4(b), we show the best observed error after a given number of function evaluations on a randomly selected run. For a particular fold, the error cannot improve unless that fold is directly queried. The algorithm makes nontrivial decisions in terms of which fold to query, steadily reducing the average error.

## 4.3 Using Small Datasets to Quickly Optimize for Large Datasets

As a final empirical analysis, we evaluate the dynamic multi-task entropy search strategy developed in Section 3.3 on two hyperparameter tuning problems. We treat the cost, $c_t(\mathbf{x})$, of a function evaluation as being the real running time of training and evaluating the machine learning algorithm with hyperparameter settings $\mathbf{x}$ on task $t$. We assume no prior knowledge about either task, their correlation, or their respective cost, but instead estimate these as the optimization progresses. In both tasks we compare using our multi-task entropy search strategy (MTBO) to optimizing the task of interest independently (STBO).

First, we revisit the logistic regression problem from Section 4.1 (Figure 3(d)) using the same experimental protocol, but rather than assuming that there is a completed optimization of the USPS data, the Bayesian optimization routine can instead dynamically query USPS as needed. Figure 5(a), shows the average time taken by either strategy to reach the values along the blue line. We see that MTBO reaches the minimum value on the validation set within 40 minutes, while STBO reaches it in 100 minutes. Figures 5(b) and 5(c) show that MTBO reaches better values significantly faster by spending more function evaluations on the related, but relatively cheaper task.

Finally we evaluate the very expensive problem of optimizing the hyperparameters of online Latent Dirichlet Allocation [30] on a large corpus of 200,000 documents. Snoek et al. [6] demonstrated that on this problem, Bayesian optimization could find better hyperparameters in significantly less time than the grid search conducted by the authors. We repeat this experiment here using the exact same grid as [6] and [30] but provide an auxiliary task involving a subset of 50,000 documents and 25 topics on the same grid. Each function evaluation on the large corpus took an average of 5.8 hours to evaluate while the smaller corpus took 2.5 hours. We performed our multi-task Bayesian optimization restricted to the same grid and compare to the results of the standard Bayesian optimization of [6] (the GP EI MCMC algorithm). In Figure 5d, we see that our MTBO strategy finds the minimum in approximately 6 *days* of computation while the STBO strategy takes 10 days. Our algorithm saves almost 4 days of computation by being able to dynamically explore the cheaper alternative task. We see in 5(f) that particularly early in the optimization, the algorithm explores the cheaper task to gather information about the expensive one.

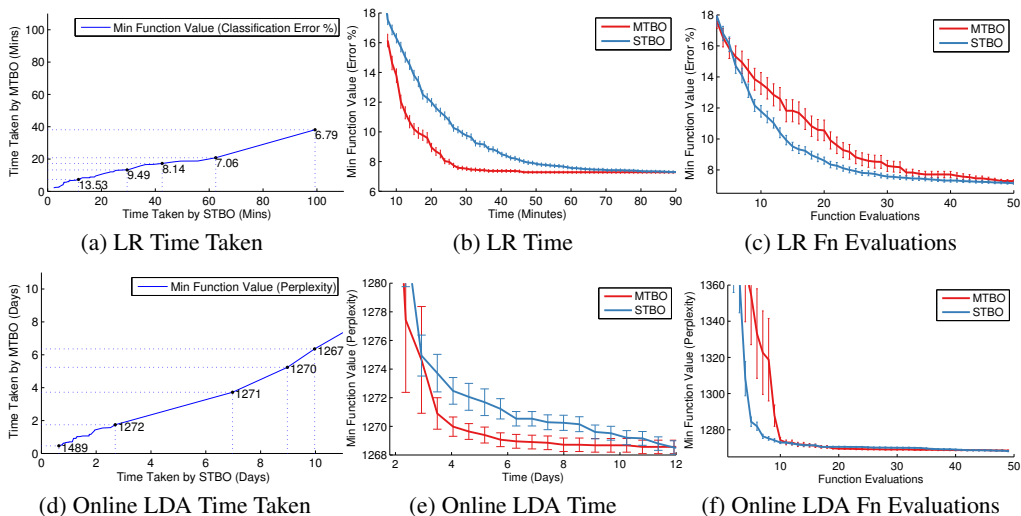

Figure 5: (a),(d) Time taken to reach a given validation error. (b),(e) Validation error as a function of time spent training the models. (c),(f) Validation error over the number of function evaluations.

# 5 Conclusion

As datasets grow larger, and models become more expensive, it has become necessary to develop new search strategies in order to find optimal hyperparameter settings as quickly as possible. Bayesian optimization has emerged a powerful framework for guiding this search. What the framework currently lacks, however, is a principled way to leverage prior knowledge gained from searches over similar domains. There is a plethora of information that can be carried over from related tasks, and taking advantage of this can result in substantial cost-savings by allowing the search to focus on regions of the hyperparameter space that are already known to be promising.

In this paper we introduced multi-task Bayesian optimization as a method to address this issue. We showed how multi-task GPs can be utilized within the existing framework in order to capture correlation between related tasks. Using this technique, we demonstrated that one can bootstrap previous searches, resulting in significantly faster optimization.

We further showed how this idea can be extended to solving multiple problems simultaneously. The first application we considered was the problem of optimizing an average score over several related tasks, motivated by the problem of $k$-fold cross-validation. Our fast cross-validation procedure obviates the need to evaluate each fold per hyperparameter query and therefore eliminates redundant and costly function evaluations.

The next application we considered employed a cost-sensitive version of the entropy search acquisition function in order to utilize a cheap auxiliary task in the minimization of an expensive primary task. Our algorithm dynamically chooses which task to evaluate, and we showed that it can substantially reduce the amount of time required to find good hyperparameter settings. This technique should prove to be useful in tuning sophisticated models on extremely large datasets.

As future work, we would like to extend this framework to multiple architectures. For example, we might want to train a one-layer neural network on one task, and a two-layer neural network on another task. This provides another avenue for utilizing one task to bootstrap another.

## Acknowledgements

The authors would like to thank Nitish Srivastava for providing help with the Deepnet package, Robert Gens for providing feature extraction code, and Richard Zemel for helpful discussions. Jasper Snoek was supported by a grant from Google. This work was funded by DARPA Young Faculty Award N66001-12-1-4219 and an Amazon AWS in Research grant.

## Footnotes

[1]Using the Cuda Convnet package: `https://code.google.com/p/cuda-convnet`

[2]Using the Deepnet package: `https://github.com/nitishsrivastava/deepnet`

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
