[Supplementary Material · nips2013transfer_supp.pdf]

# Supplementary Material for Multi-Task Bayesian Optimization

**Kevin Swersky**
Department of Computer Science
University of Toronto
kswersky@cs.toronto.edu

**Jasper Snoek**[*]
School of Engineering and Applied Sciences
Harvard University
jsnoek@seas.harvard.edu

**Ryan P. Adams**
School of Engineering and Applied Sciences
Harvard University
rpa@seas.harvard.edu

## Cold Start Experiment Details

**Logistic regression**
This experiment uses a simple logistic regression classifier taking the pixels as input and is trained with stochastic gradient descent.

| Best hyperparameter settings | | | | |
|---|---|---|---|---|
| Dataset | Learning rate | $\ell_2$ penalty | Batch size | Epochs |
| USPS | 0.0002 | 0.0032 | 560 | 161 |
| MNIST | 0.1435 | 0.0 | 206 | 685 |

**Cuda-Convnet**
We used the default architecture from the Cuda-Convnet package. This is a network that consists of two convolutional layers and two densely connected layers. When performing the transfer experiment, the epochs are scaled so that the same number of weight updates are applied to the model for both datasets. In this case, 18 epochs on SVHN is equivalent to 450 epochs on CIFAR-10. The small version of SVHN does not scale epochs, so it is only allowed to use up to 10% of the weight updates of the other models. The correlation between SVHN and CIFAR-10 was measured at $0.59 \pm 0.22$ while the correlation between SVHN and its small counterpart was measured at $0.62 \pm 0.24$.

More information on this architecture can be found on the package website: `https://code.google.com/p/cuda-convnet/.`

| Best hyperparameter settings | | | | | | |
|---|---|---|---|---|---|---|
| Dataset | Learning rate | $\ell_2$ penalty (4 layers) | Size | Scale | Pow | Epochs |
| CIFAR-10 | 0.0009 | 0.0003, 0.0075, 0.0089, 0.0028 | 3 | 0.0480 | 0.2136 | 321 |
| SVHN | 0.0044 | 0.0, 0.0, 0.0, 0.0037 | 3 | 0.0788 | 0.2700 | 18 |
| SVHN (small) | 0.0039 | 0.0, 0.0, 0.0, 0.01 | 2 | 0.1 | 0.0902 | 18 |

---

[*]Research was performed while at the University of Toronto.

**Deepnet on k-means features**

We first extract $n$ k-means features, where $n \in \{400, 1000\}$, from whitened image patches extracted from the STL-10 unsupervised image set. These are then combined using max-pooling into a $m \times m$ grid, where $m \in \{5, 7, 9\}$ resulting in a $m \times m \times n$ set of features. A convolutional neural network containing one convolutional hidden layer and one densely connected hidden layer is then applied to these. In this case, $n$ would be analogous to color channels. Each trial was allowed to use $100000$ weight updates with a batch size of $128$ and the final set of weights are used for classification. The correlation between datasets was measured at $0.5 \pm 0.28$.

The Deepnet package can be found at `https://github.com/nitishsrivastava/deepnet` and the full model specifications will be posted on the authors website.

| Best hyperparameter settings | | | | | | |
|---|---|---|---|---|---|---|
| Dataset | Learning rates | Max pooling grid size | Number of k-means features | Number of hidden units | Weight norm constraints | Dropout probabilities |
| CIFAR-10 | 0.0031, 0.0034, 0.0007 | 9x9 | 400 | 1000, 2000 | 0.25, 8.0, 2.441 | 0.7035, 0.1955, 0.4915 |
| STL-10 | 0.1, 0.1, 1e-5 | 7x7 | 1000 | 2000, 1100 | 0.25, 3.221, 0.25 | 0.5925, 0.7185, 0.9 |