[Reviews · NeurIPS 2013]

Submitted by Assigned_Reviewer_4

Update after reading the authors' rebuttal:

One of my main concerns was overhead do to GP inference. This has been addressed by the authors reasonably well. The authors acknowledge the other limitations brought up in my rebuttal. I have updated my quality score although I think the technique needs to be studied more thoroughly.

====
1. Summary


This paper presents a multi-task Bayesian optimization approach to hyper-parameter setting in machine learning models. In particular, it leverages previous work on multi-task GP learning with decomposable covariance functions and Bayesian optimization of expensive cost functions. Previous work has shown that decomposable covariance functions can be useful in multi-task regression problems (e.g. [13]) and that Bayesian optimization based on response-surfaces can also be useful for hyper-parameter tuning of machine learning algorithms [6, 7].

The paper combines the decomposable covariance assumption [13] and Bayesian optimization based on expected improvement [17] and entropy search [7] to show empirically that it is possible to :
(a) Transfer optimization knowledge across related problems, addressing e.g. the cold-start problem
(b) Optimize an aggregate of different objective functions with applications to speeding-up cross validation
(c) Use information from a smaller problem to help optimize a bigger problem faster

Positive experimental results are shown on synthetic data (Branin-Hoo function), optimizing logistic regression hyper-parameters and optimizing hyper-parameters of online LDA on real data.



2. Quality


The paper is technically sound in that it combines existing techniques that have been shown to be effective on multi-task regression problems and on single task optimization of ML algorithms. The paper does not make theoretical contributions to the field. Instead, it uses heuristics for the tasks at hand. For example, when defining a procedure for optimizing an average function over multiple tasks and when defining an acquisition function based on the entropy search for the multi-task problem.

The claims are supported empirically on various synthetic and real problems. However, the following issues are of importance:

(a) The method is not consistently evaluated in terms of time but for most experiments only the number of functions evaluations is considered. It is not explicitly shown how expensive these function evaluations are on the problems at hand and how they compare to the overhead caused by using the multi-task GP response surface methodology. This is of particular importance as sampling is done to marginalize over the kernel parameters and to compute Equation (7). In other words, total time that includes the overhead of the method should be shown across all experiments.

(b) It is unclear how general the method really is. The maximum number of function evaluations ranges from 30 up to around 200. This looks very small for general machine learning problems and as one increases the number of function evaluations the overhead of the method may be too high to be applicable.

(c) There is not a lot of analysis on why the method works. In particular, in the small-to-large experiment it may look counterintuitive that "smoothing" parameters obtained in small data may help to find good "smoothing" parameters in bigger data.

(d) It has been shown the the fixed correlation assumption in multi-task regression problems may be too strong in real-world problems (see e.g. Wilson et al (2012)). This problem may be even more drastic in optimization settings where the objective functions may be correlated differently in distinct regions of the parameter space.


(e) Unlike optimization of black box functions, in most machine learning settings, there is knowledge of the function to be optimized. This does not seem to be exploited in the proposed approach.

(f) The problem of learning "spurious" correlations, especially when having a small number of function evaluations is not addressed in the paper.

(g) In the example provided in Figure 3, it looks that by having different tasks, the predicted variance is reduced and this has the positive effect of helping finding the right point in a few number of steps. However, response-based methods are known to have problems with underestimating the variance [17]. This means that some regions of the space will be massively under-explored. The paper does not address this issue.

3. Clarity


The paper is well written and well organized and provides adequate background to the reader familiar with GPs. However the following items need consideration:

(a) It does not seem that the Kronecker structure holds for all experiments. It seems that Equation (3) should refer to a decomposable covariance function rather than a Kronecker product.

(b) There is not sufficent detail in the experiments to be reproducible. for example, to generate the results in Fig 3, it is unclear what parameters are optimized in the LR case.


4. Originality

The paper is a combination of existing ideas. There are not novel theoretical contributions but the claims are supported empirically.


5. Significance

The proposed approach is unlikely to have big impact on practitioners or researchers. There are not sufficient theoretical grounds on why or when their method can be useful. The experimental settings may be quite restrictive, for example when one requires a large number of function evaluations the overhead of the technique may be too high.


6. Additional comments

(a) line 139: "due to it's simple form" --> "due to its simple form"

(b) line 313: Should it be Figure 3 (e), (f) instead of Figure 5?




References
Wilson, A. G., Knowles, D. A., and Ghahramani, Z. (2012). Gaussian process regression networks. In ICML.
Summary: This paper presents an approach to optimization of hyper-parameter in machine learning methods that combines existing ideas on multi-task learning and Bayesian optimization. Overall, the paper does not make new theoretical contributions but it supports its claims experimentally. However, the experiments fail at analyzing how general the method really is, the potential overhead of the technique and when it is likely to work or not.

Submitted by Assigned_Reviewer_5

The paper addresses the problem of automatically selecting hyperparameters for machine learning architectures. The authors consider Bayesian optimization (BO) with Gaussian processes (GP) and adapt this framework to a multi-task setup. This setup allows the sharing of information across multiple optimization problems, and can thus, for instance, exploit knowledge acquired while optimizing the hyperparameters of a learning architecture for one dataset during the optimization of these parameters for a second dataset. They explore three variants of their scheme: (1) transferring knowledge from previously completed optimizations; (2) speeding-up cross-validation by cleverly selecting folds instead of evaluating all parameter settings on all folds; and (3) a cost-sensitive version of hyperparameter optimization that can take advantage of cheaper surrogate tasks (e.g. optimizing the parameters on subsets of the data) for solving a computationally more expensive primary problem by actively selecting the task that is likely to yield the largest improvement per unit cost (e.g. compute time).



Optimizing the hyper-parameters of machine learning architectures and algorithms is an important practical problem. The paper falls in line with several recent works that apply sequential model-based optimization (SMBO) techniques, including BO with GPs, to this end (see e.g. refs in the paper). The idea of applying ideas from multi-task learning to BO for hyper-parameter optimization is not completely novel (see the very recent paper by Bardenet et al.; ref [18]), but appears promising and has, to my knowledge, not been explored widely.

The paper is generally well written, providing a good explanation of GP BO, and of the proposed extensions. There may be a few places where clarity could be improved (see below).

The contribution of the paper is primarily that it brings together, in a sensible manner, various existing tools from the machine learning literature to adapt BO to a multi-task setup (I don't think the "ingredients" themselves are really novel). Furthermore, it proposes -- and empirically explores -- three scenarios for knowledge sharing across tasks in the context of automatic hyper-parameter optimization that are likely to be useful in practice.

Specifically, it seems to be building on the work of Snoek et al. (ref [6]), integrating the multi-task kernel of Bonilla et al. (ref [13]), and the recently proposed entropy-search (ES) acquisition function of Hennig & Schuler ([7]). It further finds a new use for the idea of cost-sensitive BO that was previously proposed in [6]. These "architectural" choices seem generally sensible (see below for some additional comments). Of the three application scenarios considered I find especially scenario (3) interesting, and it appears to be a good potential use-case for cost-sensitive BO.

I find the experimental evaluation is generally convincing: For each scenario, the authors provide at least one real-world example demonstrating that the respective multi-task approach leads to equally good or better settings of the hyper-parameters in a shorter amount of time than performing BO without taking into account the information from the related task.

In some cases additional control experiments could have been performed to provide additional baselines, test the limits of the proposed approach, and to justify and tease apart the effects of the various architectural choices (see below for details).

Overall, I think this is a well written paper that is likely to be of practical value for the ML community.






Some additional comments:

- entropy search criterion (eq. 6): I might well be mistaken but I am a bit confused by the notation, specifically the conditioning of p(f | x, \theta, {x_n, y_n}) on x. If I'm understanding correctly, p(f|...) represents the K-dimensional Gaussian distribution arising from evaluating the posterior over the Gaussian process at points in \tilde{X}. So why do you condition on x, but not on the remaining points of \tilde{X}? Couldn't you just drop x here since you are indexing f in the product term?

Also, this seems to be slightly inconsistent with eq. (7) where p(f|x) indicates that the GP is evaluated at a candidate location for an observation (which, if I'm understanding correctly, doesn't necessarily have to coincide with any of the points in \tilde{X})?

- I think [7] express your equation (7) in terms of the KL between the P^y_{min} and a uniform distribution. They comment that this is a better choice than the KL you are considering (their page 14). Maybe it'd be worth commenting on this?

- two-step heuristic in CV experiments (lines 204ff): Why is this two step approach necessary? Given that the number of folds is probably not huge couldn't you optimize the average objective with respect to an x for each task, and then choose the best x from that set? (Although maybe not much will be gained by this.)

- EI vs entropy search (ES; line 240f): You say that EI does not generalize directly to the multi-task setup while ES does. Maybe you could explain why? Couldn't you use the EI on the primary task (per unit cost) as acquisition function? After all, you seem to be using EI on the primary task anyway for the generation of candidate points at which to evaluate the KL-objective (line 261f).

- eq. 9: Is there a particular reason why you express (9) in terms of a difference in entropies and not in terms of KL as in (7)?

- Experiments in sections 4.1 / 4.2: Am I right to assume that in these experiments you always use the EI criterion as acquisition function? Would you expect ES to perform any differently in these experiments? What about the comparison between STBO and MTBO in section 4.3: how much of the difference is attributable to multi-task vs. single-task and how much to the different acquisition functions?

- I assume that for the experiments in 4.1/4.2 single-task and multi-task function evaluations take roughly the same amount of time so that there is no need to plot performance as a function of compute time?

- Experiments in section 4.1: How do the presented warm-start results compare to very simple baseline strategies for transferring knowledge from previously completed optimizations such as
> just using the best-performing hyperparameter setting found in the previous optimization
> fixing some the parameters to good values found in the previous optimization and searching only over the remaining ones
> searching only over a smaller range for all or some of the parameters based on the results of the previous optimization

- Experiments in section 4.1: Is there anything to be gained by considering the results of more than a single completed previous optimization? (E.g. for CNN on SVHN: combining information from CIFAR as well as from the subset of SVHN?) Also, how does the usefulness of information sharing depend on the number of iterations performed in the previous optimization?

- Have you considered using alternative multi-task kernels, e.g. one in which task similarity is not modeled "free-form" but depends on certain task features?

- Section 4.3: Your experiments seem to be considering a rather simple case of cost-sensitive BO in that the cost exclusively depends on the task s.t. c_t(x) = C_t (or does the cost actually depend on x?), and there are only two tasks. Have you considered other, more challenging scenarios, e.g. the possibility to vary the number of datapoints in the subsets at a finer granularity?


- y-axis labels in Fig. 3e,f seem wrong
- in eq. (8) shouldn't there be a task label for each past observation e.g. {x_n, y_n, t_n}?

- additional reference: Bergstra et al., ICML 2013: "Making a Science of Model Search: Hyperparameter OPtimization in Hundreds of Dimensions for Vision Architectures"
Summary: I think this is a well written paper that is likely to be of practical value for the ML community.

Submitted by Assigned_Reviewer_6

Overview

This paper extends GP-based on Bayesian optimization in three ways: 1) multi-task GP to address cold-start problems, 2) improving efficiency of cross-validation and 3) save optimization cost using information from a cheaper function. These extensions are well supported by corresponding experimental results.

Quality
The proposed ideas are novel, the techniques are sound and the results are informative and convincing
Here are a few questions.
1. The statement of marginalizing out GP parameters by elliptical sampling may a little misleading. The reason is that, conditioning on different number of samples, the posterior distributions of the GP parameters change and running the elliptical sampling for a longer time for convergence for each new sample or function evaluation will be too expensive. So probably the sampler is used with only a few iterations (as a search method) or fixed after a while? Could the authors give more details on this?
2. A related, natural question is why not simply using Bayesian optimization to tune GP parameters based on another GP, which is hopefully less sensitive to its parameter values compared to the first GP for the original Bayesian optimization
3. Given cubic cost of GP inference, how about using Sparse GP in case we have a a complicated function for which a lot of more function evaluations are needed? This could further save the cost.

Clarity
The paper is well written and I enjoyed reading it. At some places (e.g., discussion of the need of improving cross-validation) I felt the paper is a little repetitive and wordy.
Minor typos:
Line 308: Figure 5 -> Figure 3. And explain LR in the figure subtitle.
Reference: Line 440, 459, Bayesian -> Bayesian
Line 455 gaussian -> Gaussian

Originality
Although these are extensions of a recent work, the proposed approaches are quite novel.

Significance
I expect this paper to have a high impact because tuning hyper-parameters is a common important problem to many machine learning algorithms.
Summary: The paper uses matrix-variate GP in Bayesian optimization to address cold-start problems, improve efficiency of cross-validation and save the optimization cost by using information from a cheaper function. The proposed ideas are novel, the techniques are sound and the results are informative.
Author Feedback

Author rebuttal: We thank the reviewers for their valuable feedback.

General comments:
It is important to clarify that as it is defined, EI cannot be used directly for MTBO (sec. 4.3) because the improvement on the primary task from an auxiliary observation is always 0. To solve this issue, we introduced cost-sensitive ES. We feel that this is one of the main novel contributions of the paper.

We used the open-source code from [6] as the basis for our experiments. Like them, we only draw a few samples per iteration, warm-starting from the previous value. In general, each function evaluation takes on the order of hours while selecting the next point takes seconds to minutes, so the GP overhead is negligible. Fig. 5 includes timing results, but we will provide more. For comparison, MTBO takes around 6 processor days while the grid search in [28] takes 60-120.

One reason why our method works is that problems from similar domains tend to have similar error surfaces; humans tend to make the same assumption during manual tuning. Single-task optimization generally needs to do a lot of initial exploration. In the multi-task setting, these regions are not re-explored since bad regions tend to be bad across many different tasks.

Theoretical analysis of Bayesian optimization remains an active topic of research. However, despite the lack of theoretical proofs this approach has been shown to consistently beat popular heuristics like grid search, random search and manual tuning. We feel that previous work in this area, along with our empirical results on multiple challenging machine learning problems justifies the applicability of this approach.

Regarding community interest: As machine learning models have become more sophisticated and have more knobs to turn, there has been increasing interest in tools for hyperparameter tuning. In the last several years, such tools - and Bayesian optimization in particular - have set records on a variety of difficult machine learning benchmarks. There have been recent ICML and NIPS workshops on related topics and at least one popular Coursera ML course includes Bayesian optimization as a topic.

Comments to specific reviewers:

Reviewer 1:
Quality:
a) Fig. 5a and 5d show timing comparisons. The time spent on GP inference is negligible compared to the cost of each function evaluation. See the general comments for more details.

b) While there may be cases where more than 200 iterations are needed, we show on several challenging problems that human performance can be matched or exceeded in far fewer iterations.

c) In general, we would of course not expect them to be identical. The point is that we can leverage even weak information to improve performance. This is somewhat similar to parallel tempering MCMC methods, for example, in which smoother versions of the problem inform the difficult primary problem.

d) We agree that this is a limitation, but feel that this is a topic for future work.

e) We utilize prior knowledge when setting the boundaries of the search domain. We also used prior knowledge that e.g., all tasks use convolutional networks for object recognition (sec. 4.1), or that we are performing cross-validation (4.2). In a sense, the paper is entirely about including such prior knowledge, because the user specifies which tasks are related.

f) We marginalize out the correlations with MCMC, so we hope to be robust to this when there are few evaluations; this appears to work well.

g) It is true that this is a larger issue for Bayesian optimization. However, we feel that we avoid many of these fragility issues, and observe this empirically, by using a fully Bayesian treatment that marginalizes over all GP hyperparameters.

Clarity:
a) The Kronecker structure is used in all multi-task experiments. Kx is itself decomposable.

b) The LR experiments, among others, were reproduced from [6]. We will include these details.

Reviewer 2:
Notation - Thank you, we will certainly make this more clear.

Base distribution - We compared to using uniform as a base distribution for ES. Unlike [7] we found that the previous Pmin worked better for our problems.

Average EI - On the average objective, all tasks share the same x, so there is no way to optimize this wrt x for each task individually. We did try optimizing EI for each task individually, but our current approach worked much better.

KL - Both KL and information gain are equivalent, however we felt that cost-sensitive IG is more intuitive than cost-sensitive KL. We will clarify this.

ES in experiments - As an acquisition function, we found that ES and EI perform similarly so we used EI when we could due to it’s analytic form. In sec. 4.3 we found no way to use EI (see general comments).

Timing - Yes, the cost of the multi-task version comes from having more observations, the additional cost of this is negligible compared to the function evaluations themselves.

Baselines - We found that the best settings from one task were generally not the best settings on other tasks and could be greatly improved. We will include these results as baselines. The other suggestions are interesting, but also require more domain knowledge.

4.1 - Using more tasks would likely help, as would more observations on previous tasks. The cost would be more expensive inference.

Kt - We are currently looking into using features e.g., dataset size, for Kt.

4.3 - We only used one cost per task, but [6] considered the general case and we expect that it will work in our setting too.

Citation - Thank you, we will add this reference.

Reviewer 3:
Quality:
1. See general comments above.

2. This would be very interesting if it worked, but perhaps beyond the scope of this paper.

3. We are currently exploring sparse variants for computational efficiency.

Clarity:
We will try to make the writing more concise.